# Endotoxin Tolerance Creates Favourable Conditions for Cancer Development

**DOI:** 10.3390/cancers15205113

**Published:** 2023-10-23

**Authors:** Konkonika Roy, Henryk Mikołaj Kozłowski, Tomasz Jędrzejewski, Justyna Sobocińska, Bartosz Maciejewski, Artur Dzialuk, Sylwia Wrotek

**Affiliations:** 1Department of Immunology, Faculty of Biology and Veterinary Sciences, Nicolaus Copernicus University, 1 Lwowska Street, 87-100 Torun, Poland; 2Department of Genetics, Faculty of Biological Sciences, Kazimierz Wielki University, 10 Powstańców Wielkopolskich Ave., 85-090 Bydgoszcz, Poland

**Keywords:** cancer, endotoxin tolerance, macrophage polarization M1/M2, tumour microenvironment, immunosuppression, pro-cancerogenic conditions, cytokines

## Abstract

**Simple Summary:**

Macrophages, depending on their phenotype, can either destroy or stimulate cancer cells. Therefore, it is very important to identify the conditions under which they adopt a dangerous phenotype. Our study investigated the impact of endotoxin tolerance (ET) on macrophage behaviour and its role in cancer development. By utilizing in vitro models and diverse research methods, including examining conditioned medium effects on 3D cancer cell cultures and studying macrophage-cancer cell crosstalk, we discovered that ET-induced macrophage reprogramming leads to the release of factors that promote a cancer-favourable environment. Our findings highlight the dual nature of ET as a mechanism, potentially contributing to cancer progression. This work suggests that targeting ET could offer novel avenues for cancer prevention and treatment. To the best of our knowledge, our research group is the first to uncover this adaptive mechanism’s potential role in cancer development.

**Abstract:**

Endotoxin tolerance (ET) is an adaptive phenomenon of the immune system that protects the host from clinical complications due to repeated exposure of the body to endotoxins such as lipopolysaccharide (LPS). Since ET is an immunosuppressive mechanism in which a significant reprogramming of macrophages is observed, we hypothesized that it could influence cancer development by modifying the tumour environment. This study aimed to explore whether ET influences cancer progression by altering the tumour microenvironment. Endotoxin-tolerant macrophages (Mo_ET_) were examined for their impact on breast and colon cancer cells via direct interaction and conditioned media exposure. We characterized cancer cell behaviour by viability, clonogenic potential, motility, scratch assays, and 3D spheroidal assays. Mo_ET_-derived factors increased cancer cell viability, motility, and clonogenicity, suggesting a conducive environment for cancer development. Remarkably, despite reduced TNFα and IL-6 levels, Mo_ET_ exhibited M1 polarization. These findings uncover an ET-associated macrophage reprogramming that fosters a favourable context for cancer progression across diverse tumours. Targeting ET could emerge as a promising avenue for cancer therapy and prevention.

## 1. Introduction

Cancer is one of the leading causes of death for individuals under the age of 70 years in 112 out of 183 countries, according to the World Health Organization (WHO). The most commonly diagnosed cancers are female breast, lung, and colorectal cancers [1]. There have been many advancements in cancer therapy and treatment, such as targeted therapy and immunotherapy. However, even with such advancements in early screening and treatment, the American Cancer Society has estimated that there will be approximately 300,590 new cases of breast cancer and 153,020 new cases of colorectal cancer in the United States in 2023 [2]. While the mortality rate of cancer patients still remains high, it is important to note that the survival rate has improved over the years. For instance, a study in Italy indicates the 5-year survival rate for breast cancer has increased from 75% in the mid-1970s to 91% today. Similarly, the 5-year survival rate for colorectal cancer has increased from 51% in the mid-1970s to 66% [3,4]. To further improve the outcomes of cancer patients, novel, effective therapeutic strategies are in high demand.

Cancer development is influenced by factors that are released from both the cancer cells and the cells in the surrounding environment. In research on cancer, macrophages are the subject of many studies [5,6,7]. It has been found that not only tumour-associated macrophages (TAM) but also peripheral macrophages affect cancer growth [8]. Furthermore, it is commonly accepted that they may exhibit anti-cancer and pro-cancer properties, depending on their M1 and M2 phenotypes. Briefly, the M1 phenotype is responsible for anti-cancer effects through the production of pro-inflammatory cytokines such as tumour necrosis factor α (TNFα) and interleukin (IL-6). In contrast, M2 phenotype macrophages are reported to produce anti-inflammatory factors such as tumour growth factor (TGF) β and IL-10 and are considered a pro-cancer pool [9,10].

The main role of macrophages is to protect organisms from bacterial and viral infections by secreting antimicrobial mediators and pro-inflammatory cytokines [11]. These cells are very sensitive to lipopolysaccharide (LPS), an endotoxin derived from the outer membrane of Gram-negative bacteria that is one of the most common factors that humans and animals are exposed to throughout their lives [12]. In response to LPS, macrophages shift towards the M1 phenotype and produce pro-inflammatory factors [13,14]. However, if the macrophages are exposed to LPS for a prolonged period, a state of endotoxin tolerance (ET) develops [15]. As a result, the pro-inflammatory response of macrophages is attenuated, and fever is abolished [16]. This is believed to be a protective mechanism against the harmful effects of endotoxin-induced acute inflammation [17]. On the other hand, ET turned out to be a “double-edged sword” because it is involved in promoting secondary infections, sepsis, and eventually death [18,19,20]. 

Since reprogramming of endotoxin-tolerant macrophages adversely affects the course of infection, we wondered whether ET may also affect cancer development. Thus, in the current study, we hypothesized that ET-related immune paralysis can create conditions favourable to cancer. To test this, we first established an in vitro model of ET, and then we studied the effect of endotoxin-tolerant macrophages on cancer cell behaviour. 

Our data clearly shows that endotoxin-tolerant macrophages release factors that support tumour development and enhance cancer aggressiveness.

## 2. Materials and Methods

### 2.1. Cell Culture

The murine macrophage cell line RAW 264.7 was obtained from the European Collection of Authenticated Cell Cultures (Salisbury, UK), while the breast cancer cell line 4T1 and the colon cancer cell line CT26 were purchased from the American Type Culture Collection (Manassas, VA, USA). All the cell lines were cultured in high glucose Dulbecco’s Modified Eagle’s Medium (DMEM) supplemented with 10% fetal bovine serum (FBS), 100 μg/mL streptomycin, and 100 IU/mL penicillin (all compounds from Sigma-Aldrich, Darmstadt, Germany) and were incubated at 37 °C in a humidified atmosphere with 5% CO_2_. The cells were sub-cultured every 2–3 days. To detach the adherent 4T1 and CT26 cells, 0.25% trypsin-EDTA solution (Sigma-Aldrich) was used after the cells reached 70–80% confluency, and RAW 264.7 cells were easily detached by the light scraping.

### 2.2. Preparation of Lipopolysaccharide (LPS) Solution

Lipopolysaccharide (LPS) derived from *Escherichia coli* (0111: B4, Sigma Aldrich) was dissolved in a sterile phosphate-buffered saline (PBS). LPS was used in the experiments at a working concentration of 100 ng/mL.

### 2.3. Induction of Endotoxin Tolerance (ET) in RAW 264.7 Macrophage Cells

RAW 264.7 macrophages were seeded in a 24-well plate at a concentration of 2 × 10^5^ cells/well in 2 mL of DMEM medium supplemented with 10% FBS and pre-incubated for 24 h. The cells were then maintained in the following three conditions: non-tolerant macrophages (Mo_NT_), tolerant macrophages (Mo_ET_), or macrophages treated only once with LPS for 24 h (Mo_LPS_). Mo_LPS_ were used as a positive control. To obtain Mo_ET_ cells, RAW 264.7 cells were stimulated for 24 h with 100 ng/mL of LPS, followed by a wash with PBS and further culturing in a similar dose of LPS-containing media for another 24 h. Finally, the post-culture supernatants were collected and stored at −80 °C until the ELISA assays were performed to determine cytokine levels. 

### 2.4. Preparation of Conditioned Media from RAW 264.7 Macrophages

The RAW 264.7 cells were seeded in Petri dishes at a density of 3 × 10^6^ cells/plate in 8 mL of DMEM medium supplemented with 10% FBS and then pre-incubated for 24 h. For the induction of ET, these RAW 264.7 cells were cultured in a media containing 100 ng/mL of LPS and 1% FBS for 24 h, followed by a wash with PBS and stimulation with 100 ng/mL of LPS for another 24 h. Non-tolerant macrophages were cultured in the same condition in the absence of LPS. Finally, the cell culture supernatants were collected and centrifuged (2000× *g*, 5 min.) to remove the cell debris. The conditioned media (CM) from non-tolerant macrophages (CM_NT_) and tolerant macrophages (CM_ET_) were stored at −80 °C until needed in further experiments.

### 2.5. Cell Viability Assay

For the evaluation of 4T1 and CT26 cancer cell viability after stimulation with different concentrations of CM_NT_ and CM_ET_, a 3-(4,5-Dimethyl-2-thiazolyl)-2,5-diphenyl-2H-tetrazolium bromide (MTT; Sigma Aldrich) assay was performed. First, the cells were seeded into 96-well plates at a density of 2 × 10^3^ cells/well and pre-incubated for 24 h in the DMEM containing 10% FBS. The cells were then stimulated with CM^NT^ or CM^ET^ in 1% FBS/DMEM media at 10, 25, 50, and 75% concentration for 24 h and 48 h. After the treatment, the cells were washed with PBS and incubated in a red phenol-free culture medium containing 0.5 mg/mL MTT solution for 3 h at 37 °C. Then, to dissolve the formazan crystals, the media was aspirated, 100 μL of DMSO was added, and the plate was placed on an orbital shaker and mixed horizontally for 15 min. Lastly, the optical density was measured at the wavelength of 570 nm (with a reference wavelength of 630 nm) using a Synergy HT Multi-Mode Microplate Reader (BioTek Instruments, Winooski, VT, USA). The viability of conditioned media-treated cells is shown as the percentage of cells incubated in a complete DMEM medium containing 10, 25, 50, or 75% of culture medium supplemented with 1% FBS.

### 2.6. Colony Formation Assay

The effect of the CM_NT_ and CM_ET_ on the colony formation capacity of both the 4T1 and CT26 cells was evaluated by seeding 1 × 10^5^ cells/well in a 12-well plate and incubating overnight. The cells were then treated with different concentrations of CM_NT_ and CM_ET_ (10–50%) for 48 h. After the stimulation, the media were removed, and the cells were washed with PBS and trypsinized. These cells were then seeded at a density of 200 (4T1 cells) or 400 (CT26 cells) cells/well and maintained in the normal culture media supplemented with 10% FBS for 5 to 7 days, respectively. The fresh media were added every 2–3 days. At the end of the time point, the colonies were fixed with 100% *v*/*v* methanol for 20 min, washed once with distilled water, and then stained with 0.5% crystal violet (the solution prepared in 25% *v*/*v* methanol) solution for 20 min. After the staining, the images of the colonies from each cell line were obtained and analyzed by ImageJ software (National Institutes of Health, Bethesda, MD, USA) using the colony counter plugin. 

### 2.7. Scratch Assay

To evaluate the influence of the CM_NT_ and CM_ET_ on 4T1 and CT 26 cancer cell motility, the cells were seeded at a concentration of 3 × 10^4^ cells/well in a 24-well plate and then cultured in DMEM containing 10% FBS until 90% confluency was obtained. Scratch was made mechanically using a 10 μL pipette tip, followed by the removal of the media, and immediately washed with PBS to remove any unbound cells. The cells were stimulated with different concentrations (10–50%) of the conditioned media CM_NT_ and CM_ET_ for 20 to 24 h. Images of the scratch closure were obtained at 0 h and 20 h for 4T1 cells and at 0 h and 24 h for CT26 cells with an inverted Leica Dmi1 microscope with a digital camera (Wetzlar, Germany). These images were analyzed with the ImageJ software (National Institutes of Health, Bethesda, MD, USA) by calculating the distance between the edges of the scratch. The percentage of scratch closure was calculated by the following formula:Scratch closure (%)=(D0 – D[20 or 24])/D0×100

In this equation, *D*0 represents the distance between the edges of the wound at 0 h, and *D* [20 or 24] is the distance at 20 and 24 h, respectively.

### 2.8. 3D Spheroidal Assay

To examine the effect of the CM_NT_ and CM_ET_ on the process of spheroid formation of 4T1 and CT26 cancer cells, the spheroids were generated using the hanging drop method. The 4T1 and CT26 cells at a density of 3 × 10^4^ were cultivated on the upper lips of Petri plates in a volume of 30 µL of the different concentrations of the CM (10–50%), and cancer cell spheroids cultured in the different concentrations of the non-treated culture media (10–50%) was used as a positive control. The spheroids were maintained in the conditioned media for 48 h, and then their images were obtained using a simple microscope using 100 times magnification. These images were analyzed with the ImageJ software (National Institutes of Health, Bethesda, MD, USA) by calculating the area observed of each spheroid.

### 2.9. Co-Cultures of Cancer Cells and RAW 264.7 Macrophages

To elucidate the role of the endotoxin-tolerant microenvironment in tumourigenesis, the 4T1 and CT26 cancer cells were cultured either independently or co-cultured with Mo_NT_ and Mo_ET_ cells. Five experimental conditions were evaluated, i.e., Mo_NT_, Mo_ET_, or 4T1/CT26 cancer cells cultured independently or 4T1/CT26 cells co-cultured with Mo_NT_ or Mo_ET_. Co-culture of 4T1/CT26 cells and Mo_NT_ treated with LPS for 24 h was used as a positive control. In a common 6-well plate, the Mo_NT_ or Mo_ET_ cells were seeded at a concentration of 1 × 10^5^ cells/well together with the 4T1 or CT26 cells, which were seeded at a concentration of 3 × 10^5^ cells/well. The cells were then maintained in DMEM media supplemented with 1% FBS for 24 h. After the incubation, the cell culture supernatants were collected and stored at −80 °C until the evaluation of cytokine levels (TNFα and IL-6) by ELISA technique.

### 2.10. Analysis of Cytokine Production

The protein level of pro-inflammatory cytokines (TNFα and IL-6) in the Mo_NT_, Mo_ET_, and Mo_LPS_ post-culture and the co-culture experiment supernatants were analyzed with ELISA kits from R & D Systems (Minneapolis, MN, USA) according to the manufacturer’s protocols. Colourimetric changes in the assays were detected with Synergy HT Multi-Mode Microplate Reader (BioTek Instruments, Winooski, VT, USA). The same method was also used to assess the level of cytokines produced by the cancer cells.

### 2.11. Flow Cytometry Analysis

To identify the M1/M2 polarization phenotype of the Mo_ET_, flow cytometry was performed by staining these macrophages with fluorescein isothiocyanate (FITC)-labelled anti-CD80 MoAb and allophycocyanin (APC)-conjugated anti-CD163 MoAb (Sony Biotechnology Inc., San Jose, CA, USA). In this analysis, two experimental conditions were maintained, i.e., Mo_NT_ (control) and Mo_ET_. Cells were seeded at a density of 1 × 10^6^ cells in small flasks and pre-incubated in DMEM containing 10% FBS overnight. The culture media was removed the next day, and the cells were gently washed with PBS, followed by the first stimulation with LPS 100 ng/mL for 24 h. After 24 h, the cells were again washed with PBS and stimulated for another 24 h. At the end of the stimulation, the media was removed, the monolayer of the cells was washed once with ice-cold PBS, then again by adding another 5 mL of cold PBS, and cells were detached by gentle scraping. These cells were collected and washed with PBS three times (1300 rpm, 5 min) and incubated with Mouse Seroblock FcR (Bio-Rad, Hercules, CA, USA) for 10 min. After the initial incubation, staining with anti-CD80 and anti-CD163 antibodies was done in the dark for 30 min. To remove unbounded antibodies, three washes with PBS were performed, followed by a final resuspension in 500 μL of PBS. A BriCyte E6 flow cytometer (Mindray, Shenzhen, China) was used to perform the analysis.

### 2.12. Statistical Analysis

Statistical comparisons were conducted with the GraphPad Prism 7.0 software (GraphPad Software Inc., San Diego, CA, USA). All of the data are expressed as the mean ± standard error of the mean (SEM) of three independent experiments and analyzed using one-way ANOVA followed by Bonferroni’s multiple comparisons test with the level of significance set at *p* < 0.05.

## 3. Results

### 3.1. Endotoxin-Tolerant Macrophages Displayed Decreased Expression of TNFα and IL-6

An indicator of endotoxin tolerance (ET) is the decreased production of pro-inflammatory cytokines such as TNFα and IL-6 from macrophages during continued endotoxin exposure. Therefore, to determine the induction of ET, in this study, we evaluated the level of protein expression of TNFα and IL-6 in macrophages treated with a dose of 100 ng/mL of LPS based on the previous literature and preliminary data produced in our laboratory [21].

It is well-established that macrophages stimulated with a single dose of LPS exhibit an increase in pro-inflammatory cytokines production, which was confirmed in our results showing the upregulation of TNFα and IL-6 in the LPS-stimulated cells in comparison with non-treated cells *p* < 0.001 (Figure 1A,B, respectively). However, the continued stimulation of the cells with LPS attenuated this effect since the reduced production of both cytokines was observed in the LPS-tolerated macrophages compared with the cells stimulated with LPS only once (*p* < 0.001). 

### 3.2. Conditioned Media Derived from Endotoxin-Tolerant Macrophages Increases the Survival Capacity of Cancer Cells

To study the effects of macrophage-released mediators on 4T1 and CT26 cell viability, the cancer cells were cultured in different concentrations of RAW 264.7 cell-conditioned media (CM) derived from non-treated (CM_NT_) and LPS-tolerated cells (CM_ET_) (10%, 25%, 50% and 75%). The dose-dependent toxicity of both CM_ET_ and CM_NT_ was observed for 4T1 and CT26 cell lines (Figure 2). However, this cytotoxic effect was stronger for the cells cultivated in CM_NT_ than in CM_ET_, which was observed for CM at a concentration ranging from 25–75% after 24 (Figure 2A,C) and 48 h (Figure 2B,D) of incubation.

### 3.3. Exposure of the Cancer Cells to the Conditioned Media Derived from the Tolerant Macrophages Influences Their Clonogenic Potential 

To evaluate the effect of CM_NT_ and CM_ET_ on the colony-forming capacity (the capability of a single cancer cell to grow into a large colony through clonal expansion), we performed colony-formation assays in 4T1 breast cancer and the CT26 colon cancer cell lines. Both the cancer cell lines were treated with different concentrations of CM (10–50%) for 48 h before the seeding and culturing them in DMEM supplemented with 10% FBS. 

After 7 days, the 4T1 cells treated with the CM_ET 50%_ showed a significant increase in the number of colonies when compared to the cells treated with CM_NT 50%_ (*p* < 0.001) (Figure 3A,B). However, in the case of other concentrations of the CM_NT_ and CM_ET_, more colonies were observed in CM_NT 10%_ when compared to CM_ET 10%_ (*p* < 0.05), and no significant difference was observed in the case of CM at a concentration of 25%. 

Comparatively, the CT26 cells, after 5 days of culture, showed an increased number of colonies in the case of CM_ET 10%_ and CM_ET 25%_ when compared to the CM_NT 10%_ and CM_NT 25%_ (*p* < 0.01) (Figure 3C,D). Moreover, all the concentrations of CM_ET (10–50%)_ also showed a significant increase in the number of colonies when compared to the control cells, which were treated with DMEM containing 1% FBS for 48 h (*p* < 0.01). 

### 3.4. Conditioned Media Derived from Endotoxin-Tolerant Macrophages Increase Cancer Cell Motility

To further study the effect of the RAW 264.7 cell-conditioned media on the migration capacity of the 4T1 and CT26 cell lines, scratch assays were performed. The 4T1 and CT26 cells were cultured in CM_ET_ and CM_NT_ at different concentrations (10, 25, and 50%) for 20 and 24 h, respectively. The results showed that both 4T1 (Figure 4A,B) and CT26 cells (Figure 4C,D) cultured in CM_ET 10–50%_ demonstrated a significantly higher rate of scratch closure when compared to CM_NT 10 to 50%_ (*p* < 0.001). Interestingly, all tested concentrations of CM_ET_ also increased the cell motility when compared with the 4T1 and CT26 control cells. In contrast, the motility of 4T1 cancer cells cultivated in all tested concentrations of CM_NT_ was at the same level as the control cells, and in the case of CT26 cells, the CM_NT_ 10% showed increased cell motility when compared to the control (*p* < 0.001).

### 3.5. Conditioned Media from the Tolerant Macrophages Can Affect the Survivability of the Cancer Cells at the 3D Level 

After the evaluation of the effects of the CM_ET_ on the cancer cell monolayers, we wanted to understand whether the CM can also affect the growth of the cancer cell spheroids.

We observed that breast cancer cell spheroids treated with CM_ET 10–50%_ showed an increased area when compared to the CM_NT 10–50%_ (*p* < 0.001) and 4T1_10–50%_ (*p* < 0.001). Among tested concentrations of CM, the CM_ET 50%_ showed a significant increase in the spheroidal area after 48 h when compared to the CMNT 50%. Importantly, the cancer cell spheroids treated with CM_ET 10–50%_ showed an evidently significant increase in the area when compared to the unstimulated 4T1 cells at all concentrations (10–50%). However, this effect was reversed in the case of colon cancer cells, as the cancer cell spheroids stimulated with CM_NT 25–50%_ showed an increased growth when compared to CM_ET 25–50%_. Also, the untreated CT26 control cancer cell spheroids appeared to have significantly larger spheroidal area compared to spheroids stimulated with CM_NT 10–50%_ and CM_ET 10–50%_ conditions (*p* < 0.001) (Figure 5).

### 3.6. Endotoxin Tolerance Affects Crosstalk between Macrophages and Cancer Cells by Downregulating Expression of IL-6 and TNFα

In separate experiments, we co-cultured 4T1 and CT26 cells with Mo_ET_, Mo_NT_ or Mo_LPS_ cells to determine if direct contact with cancer cells affects endotoxin-related macrophage reprogramming of pro-inflammatory cytokines expression. The 24 h lasting co-culture of 4T1 and CT26 cells with macrophages (Mo_ET_ or Mo_NT_ or Mo_LPS_) resulted in strong stimulation of IL-6 production when compared with 4T1 and CT26 cultured alone (for both, *p* < 0.001). Additionally, we showed that ET conditions attenuated the IL-6 expression in co-cultured cells (4T1 or CT26 and Mo_ET_) compared to the IL-6 level measured in the co-culture of cancer cells with macrophages exposed previously to LPS once (i.e., for 4T1 and CT26, *p* < 0.001) (Figure 6A,C). 

We also observed a significant increase in TNFα production in both the 4T1 and CT26 cells co-cultured with Mo_LPS_, whereas prolonged stimulation with LPS that induced ET attenuated this effect. Moreover, the expression level of TNFα in the co-culture of 4T1 and Mo_NT_ was similar to that of the monoculture of Mo_ET_ and was significantly lower when compared with 4T1 cells co-cultured with Mo_ET_ (*p* < 0.001). In the case of CT26 cells, a similar pattern of expression was observed in almost all the conditions of co-culture when compared to 4T1, except in the case of CT26 cells co-cultured with Mo_ET_, which showed a significantly low expression of TNFα when compared to Mo_NT_+ 4T1 (*p* < 0.001) (Figure 6B,D). 

### 3.7. Macrophages Maintain the M1 Phenotype Even after Prolonged Exposure to LPS 

To determine whether endotoxin tolerance (ET) influences the polarization of macrophages, we assessed the expression of M1 (CD80) and M2 (CD163) polarization markers. We observed that the Mo_LPS_ and Mo_ET_ showed a statistically significant increase in the number of cells with an M1 phenotype compared with Mo_NT_ (*p* < 0.001); however, endotoxin tolerance reduced the number of M1 cells compared with Mo_LPS_ (*p* < 0.01). Additionally, there was a small but statistically significant increase in the number of cells with the M2 phenotype (*p* < 0.01 for Mo_LPS_ and *p* < 0.05 for Mo_ET_, respectively) (Figure 7).

## 4. Discussion

In the current study, we revealed that endotoxin tolerance (ET) is a novel factor that may contribute to cancer progression. ET is an adaptive phenomenon of the immune system that can occur from repeated exposure to endotoxins like LPS. Interestingly, though this immunosuppressive characteristic is considered to be a protective mechanism against endotoxin, it is identified in severe septic patients [22]. Furthermore, it has been reported that LPS-tolerant human monocytes are also hyporeactive to heat-killed *Streptococcus pyogenes*, *Staphylococcus aureus*, and *zymosan* [23]. Similarly, pre-treatment of macrophages and monocytes with other Toll-like receptors (TLRs) ligands, such as cholera toxin B chain [24] mycobacterial components such as arabinose-capped lipoarabinomannan and soluble tuberculosis factor, can lead to cross-tolerance of LPS [25]. Since experimental evidence from both animal models and clinical trials about the endotoxin utility in anti-cancer treatments has not been consistent [26,27,28], we suppose that it may have been associated with the development of endotoxin tolerance in some experimental settings. Thus, it seems plausible that ET may increase the probability of occurrence of unfavourable outcomes not only in infectious diseases but also in cancer. 

ET is characterized by a decrease in the production of pro-inflammatory cytokines such as TNFα [29,30] and IL-6 [29]. We assessed two models of tolerance in macrophages, i.e., first stimulation with LPS (in a dose of 100 ng/mL) for 24 h, followed by a second stimulation for either 6 h or 24 h. Our data proved that the second scheme (24 h + 24 h) of cell stimulation is optimal since we observed significant changes in the expression of pro-inflammatory factors (e.g., TNF-α, IL-6). This model is similar to the model used in previous studies by Nomura et al. (2000) and Xiang et al. (2009) [31,32]. Additionally, we assessed the expression of IL-1β, which in some research is also considered as the marker of ET condition [33]; however, similarly to Erroi et al. (1993) who observed only a moderate decrease in IL-1β expression in spleen homogenates of LPS tolerant mice [34], we did not notice significant changes in the expression of this cytokine between macrophages treated only once with LPS and endotoxin-tolerant macrophages. 

In our previous studies, we observed many times that macrophages are key cells involved in response to LPS [21,35]. It is commonly accepted that they may exhibit anti-cancer or pro-cancer properties, depending on their M1 and M2 phenotypes that are associated with the production of different cytokines. Although we observed a decreased expression of pro-inflammatory cytokines in Mo_ET_, these cells were polarized toward the M1 phenotype. Several studies reported that the M2 differentiation state resembles ET macrophages phenotypically [36,37] as it showed that induction of tolerance by LPS exposure of murine and human macrophages induced gene expression profiles that were consistent with M2 polarization. Nonetheless, the authors also reported that LPS re-stimulation of LPS-pre-treated macrophages resulted in sustained, rather than limited, expression of the assessed chemokines. Our results are in part parallel with Pena et al. (2011) [37], who reported that the macrophages pre-treated with LPS, which induced endotoxin tolerance, exhibited no significant change in the expression of the M2 marker CD206 despite developing tolerance with respect to pro-inflammatory cytokines. Moreover, these data are also in agreement with the findings of Rajaiah et al. (2013), who noted that the polarization of macrophages to the M2 phenotype is not clearly associated with ET [38]. Therefore, our data seem to confirm the hypothesis that the signalling pathways leading to endotoxin tolerance and differentiation of M2 are dissociable.

Although ET has been thought of as a protective mechanism against septic shock [39], the relationships between endotoxin sensitivity and carcinogenesis and protection against cancer are poorly understood. Therefore, using an experimental approach helpful in mimicking the microenvironmental conditions of the tumour, in this work, we studied the effect of endotoxin-tolerant macrophages on two different cancers, i.e., breast cancer cells and colon cancer cells. 

Our results of the viability test demonstrated that conditioned media obtained from endotoxin-tolerant macrophages exhibit a diminished dose-dependent cytotoxic effect on breast cancer cells and colon cancer cells as compared to conditioned media obtained from non-tolerant macrophages. Furthermore, these data are in accordance with our observation of colony-forming capacity. This test showed that exposure to the conditioned media obtained from the tolerant macrophages triggers an increase in the number of colonies when compared to the controls at the highest concentration in the case of both cell lines. It means that factors released by endotoxin-tolerant macrophages stimulate each single cancer cell to grow into a large colony through clonal expansion. 

Apart from characteristics proving the increased potential of cancer cells to proliferate in the environment created by Mo_ET_, we analyzed the capacity of these cells to move. The migration of the cancer cells is a pivotal step of metastasis, which is the primary cause of death for patients with solid tumours [40,41]. Similar to previous tests, we observed an enhanced migration of breast cancer and colon cancer cells when stimulated by a conditioned medium from endotoxin-tolerant macrophages in comparison to a conditioned medium from control, non-tolerant macrophages. To the best of our knowledge, the effect of conditioned media derived from endotoxin-tolerant macrophages on cancer cell motility has not been investigated so far.

To evaluate the effect of the conditioned media from endotoxin-tolerant macrophages on solid tumours, we used a 3D spheroidal cancer cell model. It is a critical issue since tumours exhibit greater resistance to treatment compared to cancer cells grown as a single layer. This occurs due to a phenomenon called multicellular resistance, which is caused by factors such as cell–cell interactions, cell–matrix interactions, and the three-dimensional structure of tissues [42,43]. After 48 h, we observed a major increase in the spheroidal area in the case of breast cells at the highest concentration of the conditioned media from tolerant macrophages, while the inverse effect was observed in the case of colon cancer. It is indicative of the variance of the effect towards types of cancer that differ with LPS exposure. As Zhu et al., 2016 reported that the LPS was responsible for promoting migration and invasion of colon cancer through VEGF-C activation but not proliferation [44]. A reason behind this can be the continuous exposure of the colon cancer cells to LPS from the intestinal bacteria, unlike breast cancer cells. These data correlate with the various literature [6,45,46], which states that the higher infiltration of the macrophages can improve the survival capacity in the case of colorectal cancer patients. Thus, our results suggest that the factors released by endotoxin-tolerant macrophages may increase at least breast cancer cell growth and survival, while colon cancer reacts differently.

Since we observed that factors released by endotoxin-tolerant macrophages may affect cancer cells, we were further interested in whether contact with cancer cells affects endotoxin-related macrophage reprogramming of pro-inflammatory cytokines expression. Therefore, we studied the direct cell-to-cell crosstalk between endotoxin-tolerant macrophages and breast cancer cells or colon cancer cells. We found that in the co-culture of endotoxin-tolerant macrophages and cancer cells, the lower level of both IL-6 and TNFα cytokines was still observed. It proves that direct crosstalk between macrophages and cancer cells does not eliminate this effect in macrophages. In particular, in the co-culture of endotoxin-tolerant macrophages and breast cancer cells, the IL-6 expression appeared to be at a level comparable to that of the monoculture of endotoxin-tolerant macrophages.

Our research is grounded in the hypothesis that the immunosuppression triggered by endotoxin tolerance (ET) in macrophages, a crucial component of an organism’s immune system, as well as within the tumour microenvironment, may foster conditions that do not inherently hinder the progression of cancer. Our approach involves investigating the impact of these tolerant macrophages on various aspects of cancer cells, such as viability, motility, clonogenic potential, and spheroid formation. Through this examination, we aim to substantiate the influence of ET on cancer development. By analyzing these findings, we propose a potential correlation between immune system impairment caused by ET and the onset of cancer (Figure 8). However, due to the intricate nature of ET as a multifaceted mechanism, more comprehensive investigations are needed. Consequently, our future objective is to conduct further in-depth studies to unravel the intricate relationship between cancer and endotoxin tolerance by studying the effect of ET on cancer in in vivo and also by examining the factors that are being secreted by these endotoxin-tolerant macrophages. Crucially, it is essential to examine how these factors impact various types of tumour models. Additionally, it is valuable to explore other aspects of cancer cell migration when influenced by ET macrophages in depth.

## 5. Conclusions

In this study, we confirmed macrophages’ susceptibility to endotoxin that changes over time, and finally, a state of endotoxin tolerance develops. Importantly, we proved for the first time that endotoxin-tolerant macrophages are reprogrammed and release factors that can affect cancer development and behaviour. Although there are many studies showing that macrophages may stimulate cancer development [47], our experiments proved that endotoxin-tolerant macrophages stimulate even more cancer-friendly conditions. Thus, in experiments on cancer cells and in clinical trials with endotoxin, it is important to monitor whether a state of endotoxin tolerance has developed. If so, it should be reversed to prevent a cancer-conductive environment. Furthermore, it is likely that many attempts to use endotoxin to cure cancer were inconclusive due to not considering the possibility of developing a state of endotoxin tolerance.

We believe that exploring the reprogramming mechanism of endotoxin tolerance may be an important factor to consider in achieving better outcomes in cancer patients. Therefore, further experiments are needed to fully understand the molecular mechanisms underlying the pro-cancer properties of ET.

## Figures and Tables

**Figure 1 cancers-15-05113-f001:**
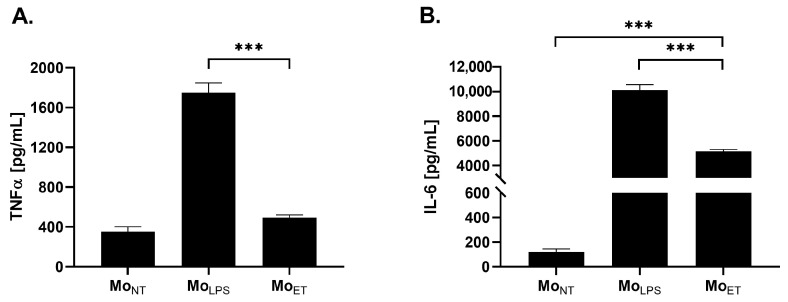
The concentration of TNFα and IL-6 produced by the following groups of RAW 264.7 cells: non-treated macrophages (Mo_NT_), LPS-tolerated macrophages (Mo_ET_), and macrophages treated with LPS only once (Mo_LPS_). Cells were stimulated either once for 24 h, Mo_LPS_ (24 h) or twice for 24 h, Mo_ET_ (48 h) with LPS at a concentration of 100 ng/mL. The amount of TNFα (**A**) and IL-6 (**B**) was assessed by ELISA. The data are shown as the means ± SEM of three independent experiments with three wells for each condition. Asterisks denote a significant difference between Mo_NT_ and Mo_LPS_ (*** *p* < 0.001).

**Figure 2 cancers-15-05113-f002:**
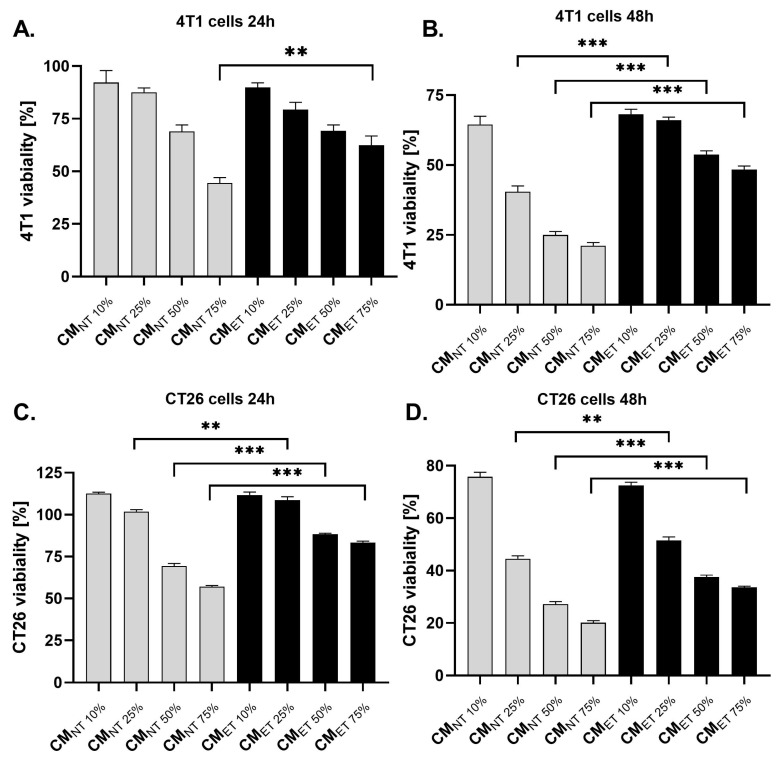
Cell viability of breast cancer 4T1 cells (**A**,**B**) and colon cancer CT26 (**C**,**D**) cells stimulated with different concentrations (10–75%) of conditioned media (CM) derived from endotoxin-tolerant macrophages (CM_ET_) and non-treated macrophages (CM_NT_) for 24 and 48 h. Cell viability was assessed by the MTT colourimetric method. The results are expressed as a percentage of control non-stimulated cells (which is represented as 100%). The data are shown as the mean ± SEM of three independent experiments with six wells for each condition. Asterisks denote significant differences between the respective concentrations of CM_ET_ and CM_NT_ (*** *p* < 0.001, ** *p* < 0.01).

**Figure 3 cancers-15-05113-f003:**
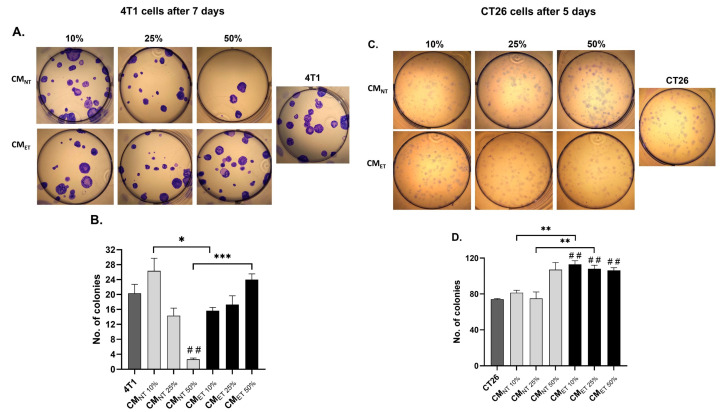
The number of colonies of 4T1 (**A**,**B**) and CT26 (**C**,**D**) cancer cells counted after 7 and 5 days of culture, respectively. Before seeding, the cells were cultivated in the conditioned media (CM) derived from endotoxin-tolerant macrophages CM_ET 10–50%_ and non-treated macrophages CM_NT 10–50%_ for 48 h. The data are shown as the mean ± SEM of three independent experiments with 3 wells for each condition. Asterisks denote a significant difference between the cells cultured in CM derived from non-treated cells (Mo_NT_) and LPS-tolerated cells (Mo_ET_) (*** *p* < 0.001; ** *p* < 0.01; * *p* < 0.05). Hashes denote the significant difference between CMET and the control cells (## *p* < 0.01).

**Figure 4 cancers-15-05113-f004:**
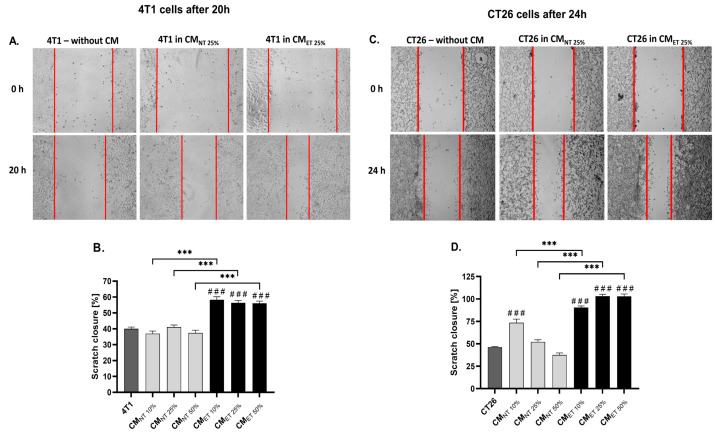
The motility of 4T1 (**A**,**B**) and CT26 (**C**,**D**) cancer cells cultured in the conditioned media (CM) derived from endotoxin-tolerant macrophages (CM_ET_) and non-treated macrophages (CM_NT_) at a concentration of 10–50% for 20 h and 24 h, respectively. Cell migration was assessed with a scratch assay. (**A**,**C**) present the representative images of the cells treated with CM at a concentration of 25%. (**B**,**D**) show the quantitative scratch closure measured between 0 h and 20 or 24 h (%) using ImageJ software. (*** *p* < 0.001). The data are shown as the mean ± SEM of three independent experiments with 3 wells for each condition. Asterisks denote a significant difference between the cells cultured in CM_NT_ and CM_ET_ (*** *p* < 0.001). Hashes denote the significant difference between MoET and the control cells (### *p* < 0.001).

**Figure 5 cancers-15-05113-f005:**
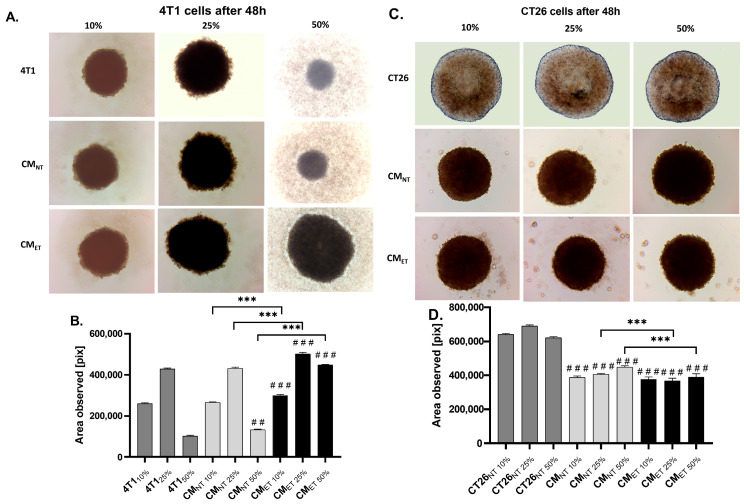
The area of 3D spheroids formed by the breast cancer 4T1 cells (**A**,**B**) and colon cancer CT26 cells (**C**,**D**) in the conditioned media (CM) derived from endotoxin-tolerant macrophages (CM_ET_), non-treated macrophages (CM_NT_), and normal media at a concentration of 10–50% after 48 h. Asterisks show a significant difference between the cells cultured in CM_ET_ and CM_NT_ (*** *p* < 0.001). Hashes denote the significant difference between Mo_ET_ and the control cells (### *p* < 0.001; ## *p* < 0.01).

**Figure 6 cancers-15-05113-f006:**
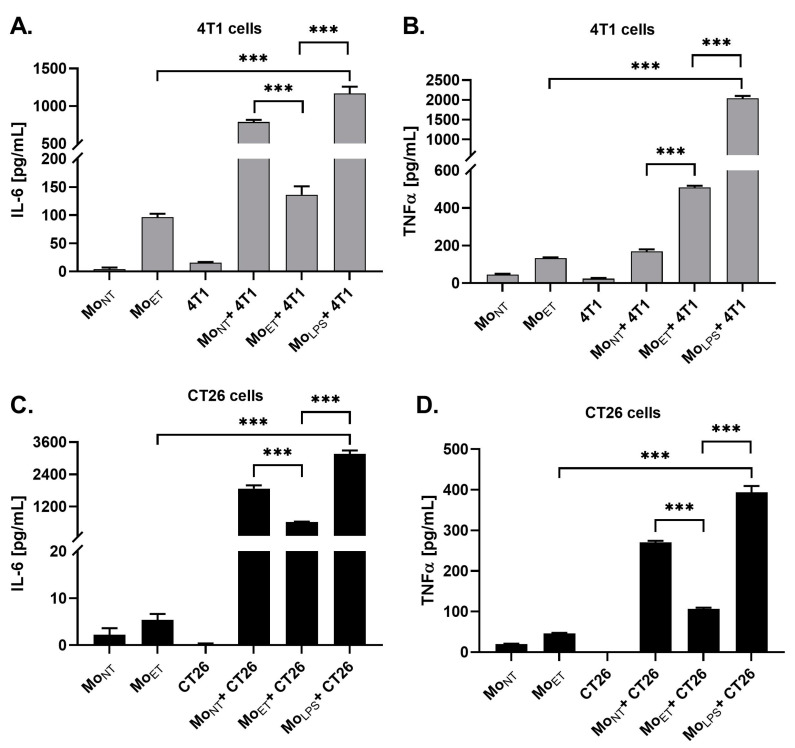
Protein concentration of IL-6 (**A**,**C**) and TNFα (**B**,**D**) produced in a monoculture of Mo_NT_, Mo_ET_, 4T1 or CT26, and co-culture of 4T1 or CT26 cells with Mo_ET_, Mo_NT_, and Mo_LPS_ that was determined by ELISA assays. Asterisks denote significant differences between the co-culture conditions of 4T1 or CT26 cells with Mo_ET_ and Mo_NT_ (*** *p* < 0.001). The data are shown as the mean ± SEM of three independent experiments with three wells for each condition.

**Figure 7 cancers-15-05113-f007:**
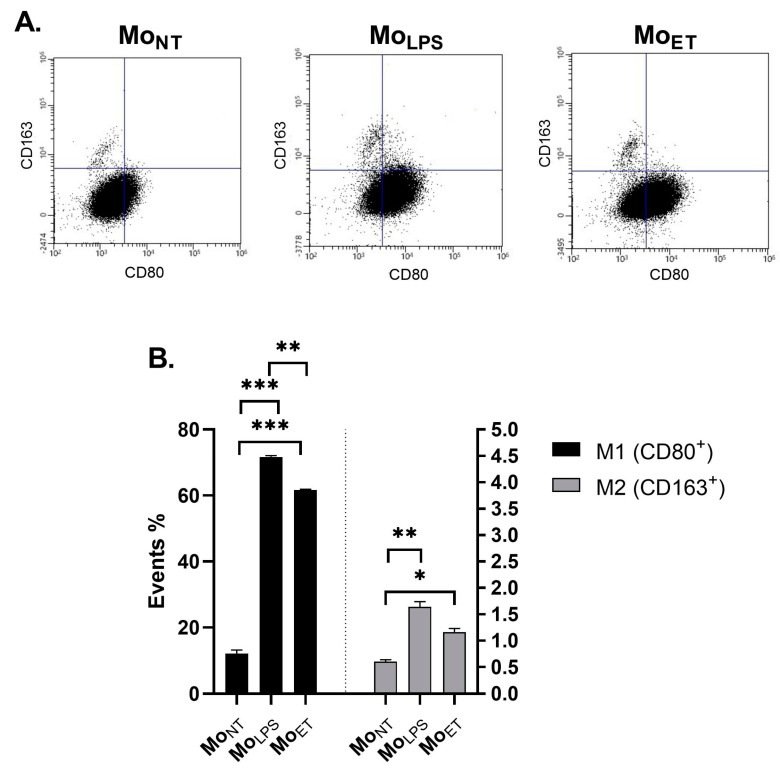
Characterization of the macrophage phenotype in LPS-tolerated RAW264.7 cell. The percentage of M1 cells is plotted on the left *Y*-axis and M2 cells on the right *Y*-axis, respectively. To induce endotoxin tolerance, the cells were stimulated twice with 100 ng/mL of LPS. The data are shown as the mean ± SEM of three independent experiments. Asterisks denote a significant difference in individual groups of cells (*** *p* < 0.001, ** *p* < 0.01, * *p* < 0.05).

**Figure 8 cancers-15-05113-f008:**
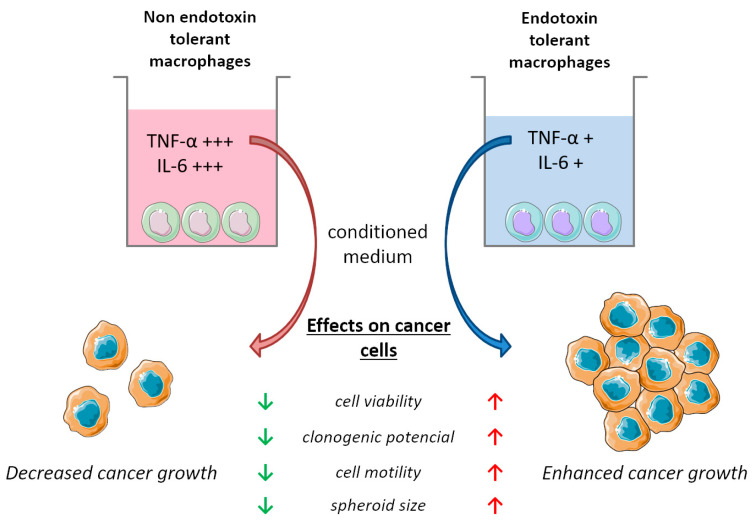
Effect of endotoxin tolerance on cancer cell growth in vitro. Endotoxin-tolerant macrophages release low levels of TNF-α and IL-6 into the medium. Conditioned medium collected from these macrophages enhances cancer aggressiveness measured by cell viability, clonogenic potential, cell motility, and spheroid size. Up and down arrows indicate the direction of effect in response to each conditioned medium. (Partly generated using Servier Medical Art, provided by Servier, licensed under a Creative Commons Attribution 3.0 unported license.)

## Data Availability

The data presented in this study are available within the article; further inquiries can be directed to the corresponding author/s.

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
