# Peer review of "Endotoxin Tolerance Creates Favourable Conditions for Cancer Development"

_cancers, 2023, doi:10.3390/cancers15205113_

Round 1
Reviewer 1 Report
The manuscript is well written, the results are clearly shown , and the methods are adequate.
I have only few minor points:
-Macrophages stimulation was conducted in 1% FBS. Usally, when LPS is used, it is important to have FBS as it contains the LBP (LPS binding protein), which binds and carries the LPS, that otherwise would be unsoluble in aqueous buffer. Why did the authors chose a percentage of FBS so low?
-The effects of treated-macrophages CM is clearly shown; however I have a concern. Did the authors checked whether this effect could be due to the presence of LPS in the medium?
Reviewer 2 Report
The manuscript of Roy et al studies dangerous phenotypes of macrophages and the impact of endotoxin tolerance on their behaviour in relation to cancer development.
The manuscript is well written and I consider it suitable for publication after minor revisions.
Figure 4 A-C. Would it be possible to have the pictures in gray scale? I would also like to ask you whether other pictures, where the width of the wound at time 0 is similar, might be used. Could the authors evaluate other parameters of single cell migration (speed, directness,...).
Reviewer 3 Report
Remarks to the Authors
In the manuscript entitled Endotoxin tolerance creates favourable conditions for cancer development, Roy et al., described the impact of endotoxin tolerance (ET) on macrophage behaviour and its role in cancer development using various in vitro approaches. The manuscript is overall well-written and the methods are properly described and therefore ready to be published.
1. What are the levels of immunosuppressive IL-10 in the MoET?
2. Do the MoET have an effect also on no tumoral cell line?
3. Would be useful to have an illustration to sum up the research highlights of the paper
Reviewer 4 Report
Major Comments:
1. How did the authors select the dosage of LPS?
2. How did the authors confirm that the CMET is free from LPS? The authors should prove that lesser cytotoxic effect of CMET on tested cell lines is solely due to the effect of LPS-treated macrophage-released mediators and not due to any direct effect of LPS contaminated in CM.
3. The authors did experiments only on two specific cell lines. Why did they comment on diverse tumors? The authors must validate their study in in vivo animal models. This is a major concern about the practical implication of the study.
4. Why did not the authors analyze other inflammatory cytokines apart from IL-6 and TNFα?
5. The authors didn’t clarify the normal status of endotoxin tolerance in general cancer patients. Then how did the authors realize that re-programming of endotoxin tolerance will be of great significance to achieve better outcomes in cancer patients?
Minor Comments:
1. Please check the manuscript for grammatical errors and typos.
Please check the manuscript for grammatical errors and typos.
Round 2
Reviewer 2 Report
I consider the manuscript suitable for publication.
Reviewer 4 Report
The manuscript is now improved significantly.
The language is fine.